



# An update on the 4D-LETKF data assimilation system for the whole neutral atmosphere

Dai Koshin[1], Kaoru Sato[1], Masashi Kohma[1], and Shingo Watanabe[2]

[1]Department of Earth Planetary Science, The University of Tokyo, Tokyo,1130033, Japan
[2]Japan Agency for Marine-Earth Science and Technology, Yokohama, 2360001, Japan

*Correspondence to*: Dai Koshin (koshin@eps.s.u-tokyo.ac.jp)

**Abstract.** The four-dimensional local ensemble transform Kalman filter (4D-LETKF) data assimilation system for the whole neutral atmosphere is updated to better represent disturbances with wave periods shorter than 1 day in the mesosphere and
lower thermosphere (MLT) region. First, incremental analysis update (IAU) filtering is introduced to reduce the generation of spurious waves arising from the insertion of the analysis updates. The IAU is better than other filtering methods, and also is commonly used for the middle atmospheric data assimilation. Second, the horizontal diffusion in the forecast model is modified to reproduce the more realistic tidal amplitudes that were observed by satellites. Third, the Sounding of the Atmosphere using Broadband Emission Radiometry (SABER) and Special Sensor Microwave Imager/Sounder (SSMIS)
observations in the stratosphere and mesosphere also are assimilated. The performance of the resultant analyses is evaluated by comparing them with the mesospheric winds from meteor radars, which are not assimilated. The representation of assimilation products is greatly improved not only for the zonal mean field but also for short-period and/or horizontally small-scale disturbances.

## 1 Introduction

The mesosphere and lower thermosphere (MLT) region is located between the lower atmosphere region of the troposphere and stratosphere, and the ionosphere, and occupies an important position that is significantly affected by, and affects, both regions (e.g., Smith, 2012). However, the means for observing the MLT region are limited compared with the lower atmosphere, and the atmospheric general circulation model (GCM) that covers the MLT region is not mature. Therefore, its dynamics are not fully elucidated yet. In contrast to the troposphere and stratosphere where large-scale, long-period
geostrophic motions are dominant, ageostrophic motions such as small-scale, short-period gravity waves and large-scale, short-period tidal waves are relatively important in the MLT (e.g., Shepherd et al., 2000). There also make the dynamics of the MLT region difficult to study.

In the troposphere and stratosphere, diabatic heating at low latitudes and synoptic-scale waves and planetary-scale Rossby waves at mid and high latitudes play predominant roles in the Lagrangian circulation from the tropical region to both
polar regions, but in the MLT region, gravity waves are the main driver of the unique summer-to-winter pole circulation (e.g.,





Plumb, 2002). Recently, observational and model studies have reported the existence of interhemispheric coupling, namely, teleconnection between the winter hemisphere stratosphere and the summer hemisphere mesosphere (e.g., Karlsson et al., 2009; Gumbel and Karlsson, 2011). It is considered that the coupling is due to the change in the Lagrangian mean circulation in the MLT region which is caused by the modulation of gravity waves originating from the troposphere and propagating

upward through the interaction with the mean wind (Körnich and Becker, 2010). However, the details are still unknown. In addition to gravity waves, tidal waves are also dominant in the mid–low latitude MLT region. Tidal waves themselves contribute to the momentum budget in the MLT region and also modulate gravity wave momentum deposition (e.g., Fritts and Vincent, 1987; Becker, 2012; Watanabe and Miyahara, 2009). Moreover, recent studies indicate that Rossby waves are generated because of baroclinity and barotropic instability in the stratosphere and mesosphere caused by gravity wave drag

(Watanabe et al., 2009; Ern et al. 2013; Sato and Nomoto, 2015), and that secondary gravity waves are generated because of momentum deposition and/or shear instability of the mean wind cause by the primary gravity waves from the troposphere (e.g., Sato et al., 2018; Vadas et al., 2018; Yasui et al., 2018). The redistribution of momentum and energy from these waves may influence interhemispheric coupling. Therefore, global grid data covering the region from the ground to the lower thermosphere is required to study the dynamics in the MLT region and the momentum budget of the global neutral

atmosphere to elucidate the teleconnection through the MLT region.

    Global analysis data for the whole neutral atmosphere, including the MLT region, are currently very limited. This is partly due to the shortage of observational data in the MLT region. Moreover, GCMs that include the MLT region are not very mature. As stated, gravity waves, which play a crucial role in the MLT region, are usually sub-grid scale phenomena even in state-of-the-art models. Thus, they need to be parameterized in the model. However, current gravity wave

parameterizations are not perfect, particularly in the MLT region (Geller et al., 2013). Most gravity wave parameterizations assume only vertical propagation, although lateral propagation of gravity waves before reaching the MLT is significant (e.g., Sato et al., 2009; Thurairajah et al., 2020).

    Most reanalysis products released over recent years cover the pressure levels up to 0.1 hPa in the lower mesosphere over tens of years. In contrast, only a limited number of groups have developed assimilation systems which cover the whole

neutral atmosphere up to ~100 km for the purpose of analyzing specific atmospheric events. For example, McCormack et al. (2017) performed numerical simulations for two boreal winters with the high-altitude Navy Global Environmental Model (NAVGEM; Hogan et al., 2014) coupled with a hybrid four-dimensional variational scheme (4D-Var) data assimilation system, and showed that the simulated mesospheric horizontal winds reproduced the amplitude and phase of semi-diurnal variations observed by the meteor radars. Using the Canadian Middle Atmosphere Model Data Assimilation System

(CMAM-DAS; Polavarapu et al, 2005), which assimilates meteorological observations below 1 hPa, Xu et al. (2011a; 2011b) compared the analysis data with independent observations including medium-frequency and meteor radar observations for several years. They showed that the CMAM-DAS roughly captured the variability of the observed mean horizontal winds and amplitudes of tides in the mesosphere. Pedatella et al. (2018) applied the Data Assimilation Research Testbed (DART; Anderson et al., 2009) ensemble adjustment Kalman filter (EAKF; Anderson, 2001) to the Whole


Atmosphere Community Climate Model eXtended version (WACCMX; Liu et al., 2018), and investigated stratospheric sudden warming in 2009 based on a series of ensemble hindcasts initialized from the WACCM+DART analysis. Koshin et al. (2020) (hereafter referred to as KSMW20) developed a data assimilation system (hereafter called Japanese Atmospheric GCM for Upper Atmosphere Research-Data Assimilation System; JAGUAR-DAS) with a four-dimensional Local Ensemble Transform Kalman Filter (4D-LETKF; Miyoshi and Yamane, 2007) using the JAGUAR (Watanabe and Miyahara, 2009).

The first version of JAGUAR-DAS by KSMW20 assimilated satellite temperature data from the Aura Microwave Limb Sounder (MLS; Livesey et al., 2020) as well as a conventional observation dataset. They confirmed that the time variation of obtained horizontal winds with periods longer than several days was consistent with the radar observations in the upper mesosphere. It should be noted that the global data for the whole neutral atmosphere by assimilation from these previous studies have been produced for a couple of years at most, and that they are generally not available to the public.

75        In this study, we update the data assimilation system developed by KSMW20, particularly to better reproduce high-frequency fluctuations, including atmospheric tides. The changes from the previous system are as follows: first, the Incremental Analysis Updating (IAU) process is introduced as a filtering method to suppress spurious waves generated by the assimilation increment. Second, the horizontal diffusion is modified to reproduce realistic tidal wave amplitudes. Third, non-sun-synchronous satellite observations by the Thermosphere Ionosphere Mesosphere Energetics and Dynamics

(TIMED) Sounding of the Atmosphere using Broadband Emission Radiometry (SABER, Remsberg et al., 2008) and sun-synchronous satellite observations at different local times by the Defense Meteorological Satellite Program (DMSP) Special Sensor Microwave Imager/Sounder (SSMIS, Swadley et al., 2008) are assimilated in addition to the Aura MLS which has a sun-synchronous orbit and conventional dataset.

        The analysis increments in assimilations correct the model variables to get closer to the assimilated observations.

However, corrected variables from assimilations do not necessarily obey the model equations. Thus, large analysis increments sometimes act to generate spurious high-frequency waves. A forecast initialized by analysis that has been contaminated with spurious waves can lead to unphysical states or model failure (e.g., Sankey et al., 2007). It is difficult to separate the waves in the real atmosphere from the spurious waves. Also, spurious waves arising from the insertion of the analysis updates may be more problematic for data assimilation for the middle and upper atmosphere. The model bias is

often large in the middle atmosphere compared with the lower atmosphere, which results in large analysis increments, e.g., the increments of ~10 K in temperature (Hoppel et al., 2008) and ~20 m s$^{-1}$ in horizontal winds can appear. Furthermore, spurious waves generated in the lower atmosphere and propagated upward will be amplified because density decreases with altitude. Since the number of observations in the middle atmosphere is smaller than that in the troposphere, the spurious waves and the model fields disturbed by the waves are unlikely to be corrected efficiently at a later assimilation step.

Pedatella et al. (2018) assimilated the MLS and SABER observations in the middle atmosphere and pointed out that, as a result of the analysis increments, unrealistic small-scale waves appeared in the mesosphere data which could lead to the failure of model calculations. More importantly, they noted that the spurious small-scale waves cause unrealistic mixing in the lower thermosphere and have a significant influence on chemical processes. This implies that forcing from spurious





waves may contaminate the momentum budget in the MLT in the analysis data. Thus, reducing the spurious components of
the increments improves not only the wave fields but also the momentum balance in the MLT of the analysis data.

To reduce the generation of spurious waves, various methods, mainly for numerical weather prediction (NWP) of the troposphere, have been developed so far, such as normal mode initialization (NMI), digital filter (DF), and incremental analysis updates (IAU; Bloom et al., 1996) (Kalnay, 2002). An IAU with time-varying coefficients was also proposed by Polavarapu et al. (2004) and is sometimes called the 4-dimensional IAU (4D-IAU).

There are several studies looking at how to suppress spurious waves in the mesosphere by introducing filtering methods (Polavarapu et al., 2005; Sankey et al., 2007; Wang et al., 2011; Eckermann et al., 2018), while Pedatella et al. (2018) applied additional second-order divergence damping to attenuate these waves. Sankey et al. (2007) compared the DF, Incremental DF (IDF), IAU, and 4D-IAU methods from the viewpoints of wavenumber spectra and amplitudes of mesospheric tides. They concluded that the IAU is the best filtering method to reduce the spurious waves that are generated
in the troposphere and stratosphere and then propagated into the mesosphere. They also pointed out that incremental filters preserve many of the high-frequency waves in the forecast model compared with other filters which are applied to the full analysis. It is noticeable that other filtering methods such as NMI and DF not only reduce spurious waves but can excessively smooth tides and gravity waves in the forecast model. Wang et al. (2011) implemented the IAU to avoid excessive damping of the tidal waves in the upper atmosphere. Note that MERRA (Rienecker et al., 2011) and MERRA-2
(Gelaro et al., 2017) use IAU. Thus, in the present study, the IAU is used to filter the spurious waves. The IAU has a lower computational cost than the DF and IDF.

The structure of this paper is as follows. Section 2 describes the changes in the updated system. Section 3 shows how the updates affect the analysis. The target time period for this study is from January to February 2017, which is the same setting as Koshin et al. (2020). Section 4 contains a summary and concluding remarks.

## 2 Methodology

### 2.1 Data assimilation system developed by Koshin et al. (2020)

We improved the data assimilation system using the 4D-LETKF with a GCM with a top in the lower thermosphere as a forecast model developed by KSMW20. The forecast model has 124 vertical layers from the surface to ~150 km and a T42 horizontal resolution (a latitudinal interval of 2.8125°). The monthly ozone mixing ratio climatology from the United
Kingdom Universities Global Atmospheric Modeling Programme (UGAMP; Li and Shine, 1999) and monthly sea surface temperature and sea ice concentration from the Met Office Hadley Centre sea ice and sea surface temperature dataset (HadISST; Rayner et al., 2003) are linearly interpolated in time and used as boundary conditions.

For the KSMW20 system, the assimilated observation datasets are the MLS (v.4.2) temperature, which covers the whole stratosphere and mesosphere, and the National Centers for Environmental Prediction (NCEP) PREPBUFR, which is a
standard dataset for the troposphere and lower stratosphere. Bias correction and averaging that reduces the observational





resolution comparable to the forecast model resolution (super-observation) for the MLS data are performed before the assimilation (KSMW20). The PREPBUFR global observation dataset is compiled by NCEP and archived at the University Corporation for Atmospheric Research (https://rda.ucar.edu/datasets/ds337.0/). This dataset includes temperature, wind humidity, and surface pressure from radiosondes, aircrafts, wind profilers, and satellites.

KSMW20 performed a series of sensitivity tests to optimize the data assimilation parameters for the system with 30 ensemble members, such as the degree of gross error check, localization length, inflation factor, and assimilation window. The results are summarized in Table 1. In the present study, the assimilation system with the optimized parameters by KSMW20 is improved.

### 2.2 Incremental Analysis Update (IAU)

The IAU is one of the data insertion schemes used during analysis updates (Bloom et al., 1996). For the IAU, the increments are divided into a small fraction and added at each time step for a finite time period. According to Bloom et al. (1996), the IAU filtering properties are better than those of nudging schemes because the IAU has a sharper response function with less phase distortion.

In our data assimilation cycle, the analysis increments are calculated at t=00:00, 06:00, 12:00, and 18:00 UTC,
using the forecasts and observations for the time period from t−3 h to t+3 h. The increments for temperature, zonal and meridional winds, specific humidity, and surface pressure are added as forcing terms to the model equations at each time step for the time period from t−3 h to t+3 h. Finally, the results of the subsequent six-hour forecast without the IAU forcing are used to calculate the assimilation at the next analysis step. Since this method requires an additional three-hour forecast from t−3 h to t h, the resulting forecast time increases by a factor of 4/3. Note that, in our data assimilation system, the
assimilation module consumes the calculation time about 10 times as much as that of the model forecast, which means that there is little time increase for the whole analysis cycle from introducing the IAU. The performance of the IAU is examined by comparing the results with and without IAU. The latter is the same as the "Ctrl" setting in KSMW20. KSMW20 focused on relatively slowly varying components, i.e., components with time scales longer than days. In this study, so as to express tides with realistic amplitudes in the assimilation system, the horizontal diffusion of the forecast model is tuned in addition to
the IAU inclusion.

### 2.3 SABER

The SABER instrument onboard the TIMED satellite was launched in 2001. This satellite is not in a sun-synchronous orbit, and hence the local time of the measurements is not constant, which is one of the major differences from the MLS. About every 60 days, the satellite performs yaw maneuvers so that data for the region 53° S–83° N and 83° S–53° N are alternately
obtained every 60 days. Temperature data retrieved from $CO_2$ infrared limb radiance (Remsberg et al., 2008) are used for the assimilation in our system. We used version 2.0 data. The data are distributed in the altitude range from about 15 to 110 km at ~1 km intervals. The measurement uncertainty (available from http://saber.gats-inc.com/temp_errors.php) is linearly





interpolated in the vertical direction and used as observational errors in the assimilation. For example, the uncertainty values are 1.3 K at the altitude of 20 km, 2.0 K at 60 km, and 10.5 K at 100 km. Similar to the assimilation of the MLS temperature

data in KSMW20, the observations are horizontally averaged for the along-track direction to reduce the resolution comparable to the forecast model resolution before the assimilation.

### 2.4 SSMIS

The SSMIS instrument measures Earth's radiation in 24 microwave channels using a conical scan cycle with a swath width of ~1700 km (Swadley et al., 2008). One SSMIS sensor on the DMSP F17 satellite is currently in operation, although four

SSMIS instruments on DMSP satellites (F16, F17, F18, and F19) were launched. The brightness temperatures from six upper air sounding channels in a Unified Pre-Processing package (UPP) data are used for the assimilation. These channels measure the 60 GHz molecular oxygen absorption band, which is sensitive to temperatures in the upper stratosphere and mesosphere. The noise equivalent delta temperature for each channel (available from: https://directory.eoportal.org/web/eoportal/satellite-missions/d/dmsp-block-5d) is used as the observational error in our data assimilation system. The horizontal distribution of

the SSMIS observation data is denser than the model resolution. To reduce the computational cost, the observations are thinned by taking one of every 10th consecutive data points for both the along and cross-track directions.

To assimilate the brightness temperature, we implement an observation operator for the SSMIS brightness temperature. The observation operator that converts the model variables to brightness temperatures with a radiative transfer model (RTTOV v.11.3; Saunders et al., 2018) is used, which was originally developed by Teresaki and Miyoshi (2017). The

satellite radiances may include two kinds of biases: airmass-dependent and scan-dependent biases (e.g., Miyoshi et al., 2010). The airmass bias is responsible for inaccuracies in the radiative transfer calculations, which are correlated with predictors computed from the model variables. In the present study, the airmass bias is subtracted from the observed radiances following Terasaki and Miyoshi (2017). This correction relies on a linear combination of a set of state-dependent predictors including lapse rate and surface temperature. The coefficients of the predictors are estimated with the ensemble-based

variational bias correction method (VarBC; Miyoshi et al., 2010). The scan bias comes from viewing the angles of the field of view. Since the viewing angle is constant for the SSMIS because of the conical scan pattern, there is no need to take the scan bias into consideration for the present assimilation system.

### 2.5 Independent data

The zonal winds obtained from the assimilation experiments are compared with observations by meteor radars at

Longyearbyen (78.2° N, 16.0° E; Hall et al., 2002), Kototabang (0.2° S, 100.3° E; Batubara et al., 2011), and Davis Station (68.6° S, 78.9° E; Murphy et al., 2017). These radar observations are not assimilated and, thus, can be used for validation as independent reference data. Table 2 gives a brief description of these data. In the following comparison, the data averaged for the height range of 80–88 km were used. We obtained qualitatively similar results also for the meridional winds, although they are not shown.



## 3 Results

To examine the impacts of these changes in the assimilation system, several assimilation experiments for the whole neutral atmosphere up to the lower thermosphere are performed for the time period of 10 January through 28 February 2017. This period is the same as that focused on by KSMW20. We follow the experiment settings in KSMW20 including the spin-up time and the assimilation parameters. Table 2 summarizes the experiments that we performed. The assimilation system developed by KSMW20 and the analysis data calculated from the system are called as the KSMW system and KSMW analysis, respectively. The experiment with the system adopting the IAU filtering is called Expt. I. The comparison between KSMW and Expt. I shows the impact of the filtering on the reduction of spurious waves generated from analysis increments. The experiment using the forecast model with a tuned horizontal diffusion in addition to the IAU to improve the reproductivity of the tidal amplitudes in the analysis is called Expt. II. In Expt. III, the SABER observations are additionally assimilated for the same setting as Expt. II. The system in Expt. IV assimilates the SSMIS observations in addition to the Expt. III setting. Expt. IV is regarded as the new assimilation system developed in the present study.

### 3.1 Introducing IAU to the KSMW system

First, the analysis produced by the KSMW system with the IAU (Expt. I) is compared with the original KSMW analysis. Figure 1 shows longitude-latitude sections of the geopotential height anomaly from the zonal mean at 0.1 and 10 hPa at 00 UTC on 20 January 2017. The MERRA-2 reanalysis, which also adopts the IAU filtering, is shown for comparison (Fig. 1c and 1f). Disturbances with small scales of about 1000 km are conspicuous in the original analysis at 0.1 hPa (Fig. 1a), while these disturbances are not obvious in the analysis with the IAU (Fig. 1b) or the MERRA-2 reanalysis (Fig. 1c). At 10 hPa, analyses both with (Fig. 1e) and without the IAU filtering (Fig. 1d) show a wave-2 pattern in the extratropics of the Northern Hemisphere causing a split of the polar vortex, although weak smaller-scale disturbances are present only in the original KSMW analysis (Fig. 1d). It should be noted that a free-running model simulation with the same initial condition, which inherently has no analysis increments, does not show such small-scale structures (not shown). Thus, most of the small-scale disturbances in the original analysis are likely due to spurious waves generated from the analysis increments. The small-scale waves are reduced through the IAU and the resultant analysis product is similar to the MERRA-2 reanalysis (Fig. 1f).

The amplitudes of small-scale waves are relatively large at the higher altitudes. This is partly due to potentially large increments because of a model bias in the MLT region and partly due to the amplification of the spurious waves propagating from below due to the exponentially decreasing air density. The IAU suppresses the generation of the spurious waves by reducing the increment at a time step at all heights, and hence effectively suppresses both spurious waves caused in situ and those originating from lower altitudes. It is worth noting here that we conducted sensitivity experiments in which assimilation parameters including the ensemble size and the inflation factor of observation errors were tuned. However, these parameter tunings have little impact on the reduction of the spurious waves compared with the IAU.



### 3.2 Tuning of the horizontal diffusion in the GCM

KSMW20 shows good agreement between the KSMW analysis and independent observations from meteor radars for fluctuations with periods longer than several days. However, there are some discrepancies in amplitudes and phases for fluctuations with periods shorter than one day. Figures 2a–2c show the time series of zonal winds observed by meteor radars

at three stations of Longyearbyen, Kototabang, and Davis Station (black curves) and corresponding data from the KSMW analysis (red curves). At Longyearbyen and Kototabang, the dominant wave periods are about 12 and 24 hours, respectively (left and middle panels). At Davis, two kinds of fluctuations with periods of about 12 hours and about two days seem dominant (right panel). The KSMW analysis captures the relatively long-period variations at Kototabang, but significant differences are observed for fluctuations with periods shorter than one day. This is the case also for the other two stations.

The correlation coefficients between the radar and KSMW analysis time series are small at all stations.

Figures 2d–2f show the zonal wind fluctuations obtained by the KSMW system with the IAU (Expt. I). The correlation coefficients between the time series of the analysis and observations increase from 0.23 to 0.43 at Kototabang, while they are low at Longyearbyen and Davis. It is notable that the amplitudes of the short-period variation of the analysis with the IAU became significantly underestimated at all stations. The ratio (k) of the wind variance for the analysis to that

for the observation is calculated at each station. It is found that the variance in the analysis is in a range of about 40–70 % of that in the observation at all stations. We confirmed that the variance for the meridional wind fluctuations in the analysis is also small compared with radar observations at each station. Thus, the realistic amplitudes of short-period fluctuations obtained by the KSMW analysis (Fig. 2a–2c) could be accidental. Therefore, the amplitude of the tides, which are the main components of the short-period fluctuations in the MLT region, in the forecast model is examined.

Figure 3a shows the meridional wind amplitude of the migrating diurnal tide (DW1) extracted from the free-run simulation with the forecast model using a space-time spectral analysis. The tidal amplitude has broad maxima at altitudes between 70 and 90 km at latitudes of about 20° N and 20° S. The peak amplitude is about 20 m s$^{-1}$, which is roughly half of the observation by the Wind Imaging Interferometer (WINDII) (e.g., McLandress et al., 1996) and also half of that simulated by GCMs (e.g., Watanabe and Miyahara, 2009).

The KSMW system and Expt. I use the numerical model with a fourth-order (i.e., $\nabla^4$) horizontal hyperdiffusion. To obtain more realistic tidal amplitudes, the horizontal hyperdiffusion is changed to an eighth-order (i.e., $\nabla^8$) and the time constant is decreased. This diffusion acts to reduce the amplitude of large-scale (small-wavenumber) disturbances while maintaining a large amplitude for small-scale (large-wavenumber) disturbances. The e-folding time of the horizontal diffusion as a function of the wavenumber for each setting is shown in the supplement. Figure 3b shows the tidal amplitude

calculated from the free-run simulation with the eighth-order diffusion. The peak amplitude is 28 m s$^{-1}$, which is comparable to that shown by McLandress et al. (1996) and Watanabe and Miyahara (2009). Figures 2g–2i show the results of the assimilation with the IAU filtering using the forecast model with the tuned eighth-order diffusion (Expt. II). The results from Expt. II show similar time variations to the radar observations, particularly for the short-period fluctuations. Owing to the



improved reproducibility of the semidiurnal variation, the correlation coefficient between the analysis and the radar observation significantly increases from less than 0.1 to 0.6 (0.3) at Longyearbyen (Davis). Note that the ratio of the variances k at Davis is almost one, whereas those at Longyearbyen and Kototabang are slightly larger than one.

It is also interesting that the introduction of the IAU changes the time mean zonal wind at Davis. The original KSMW analysis shows a negative bias in the zonal wind compared with the observation at Davis (Fig. 2c). The bias for the analyses from experiments using the IAU (Fig. 2f and 2i) is significantly reduced. As will be shown later, this is likely
related to the reproducibility of wave forcing in the MLT region.

### 3.3 Assimilating SABER and SSMIS observations

Last, we add data for the assimilation from non-sun-synchronous satellites at local times which are different from those of MLS. Figures 3j–3l (3m–3o) show the time series from the analysis, in which the SABER observation is (SABER and SSMIS observations are) assimilated with the IAU using the forecast model with the tuned eighth-order diffusion allowing
the production of realistic tidal amplitudes [Expt. III (Expt. IV)]. The assimilation of the additional observations increases the correlation coefficients at all stations, particularly for Davis. At Longyearbyen and Kototabang, the variances of the time series become closer to those of the observations by assimilating the SABER and SSMIS data. Values of k are obtained within a range from 1.1 to 1.5 for Expt. IV. At Longyearbyen, the performance of the assimilation is best in Expt. IV in terms of k while the correlation coefficients are comparable for Expts. II, III, and IV. At Kototabang, Expt. IV shows the
best correlation to the radar observations, although the difference in k among Expts. II, III, and IV is not significant. At Davis, the improvement of the correlation by the assimilation of the SSMIS observation is larger than at the other two stations. This is because there is no SABER observations in the southern hemisphere polar region during the present analysis period. Thus, the assimilation of SSMIS observation is more important at Davis in the Antarctic. These results indicate that the constraint from assimilating the observations at different local times is crucial for reproducing tides with realistic
amplitude variation.

### 3.4 Comparison to the KSMW analysis

In this section, results from Expt. IV (hereafter referred to as the new analysis) are compared in detail with those from KSMW analysis to examine the performance of the new assimilation system. Figures 4a, 4b, 4e, and 4f show the zonal mean temperature and zonal wind from respective analyses averaged for the time period of 15 January to 20 February 2017. It is
seen that the temperature and zonal wind below ~10 hPa exhibit only slight differences. In the new analysis, the easterly jet in the summer upper stratosphere is strong and its core shifts equatorward compared with the KSMW analysis. Also, there are significant differences in the zonal wind in the MLT region, such as the height of the zero-wind layer. As a reference, the zonal mean temperature and zonal wind estimated using the geopotential height under the assumption of the gradient wind balance with the temperature from the MLS and SABER observations are also shown (Fig. 4c, 4d, 4g, and 4h). The gradient
wind cannot be estimated at the equator where the Coriolis parameter is zero, so the zonal wind at the equator is obtained



from those at 10° S and 10° N by a linear interpolation. The jet in the summer upper stratosphere and mesosphere in the new analysis seems close to the gradient wind from the SABER. Here it is worth noting that the gradient winds do not necessarily provide a good approximation of the zonal wind in the low latitudes above a height of ~80 km where the tidal wave contamination is significant to the data from satellites taking sun-synchronous orbits (Lieberman, 1999) and where the tidal

wave forcing in the meridional direction is not negligible (Miyahara et al., 2000).

Figure 5 shows longitude-latitude sections of the zonal wind and zonal gradient wind at 10, 1, 0.1, and 0.01 hPa (from the bottom) averaged for 10–20 February 2017. The zonal winds from the KSMW analysis at 10, 1, and 0.1 hPa have large-amplitude small-scale structures, which are likely due to the spurious waves generated by the analysis increments. In contrast, the new analysis shows a smoother horizontal distribution of the zonal wind at all levels. These results reflect the

application of the IAU filtering and the assimilation of additional data from the SABER and SSMIS observations. The difference in the large-scale structure between the KSMW and the new analyses is evident at 0.01 hPa. The polar night jet structure in the northern midlatitudes at 0.01 hPa in the new analysis is more similar to those of the gradient winds from MLS and SABER. The easterly winds at low latitudes for the new analysis, which are stronger than those in the KSMW analysis, are close to the gradient wind from SABER. Note that the non-zonal structure of SABER winds at low latitudes are

mainly because of the non-uniform location and local time of the orbits (not shown).

Figure 6 shows the meridional-cross sections of the zonal mean zonal wind, Eliassen-Palm (E-P) flux, and its divergence averaged for the time period of 15 January to 20 February 2017. Similar to the zonal wind and temperature, only slight differences of the E-P flux and wave forcing between the KSMW and new analyses are seen below ~10 hPa. The E-P flux divergence from the KSMW analysis shows a patchy structure in high latitude regions from 10 to 0.1 hPa (Fig. 6a). This

is primarily caused by the small-scale spurious waves generated by the increments (Fig. 1a). The patchy structure mostly disappears in the new analysis (Fig. 6b).

The absolute value of the wave forcing of the new analysis in the MLT region is more than twice that of the KSMW analysis. To examine the cause of the difference in the wave forcing in the MLT region, the E-P flux for migrating tides are analyzed. Figures 6c and 6d show the E-P flux for the migrating solar tides with zonal wavenumbers of s=1–4. A relatively

large difference in wave forcing because of the migrating tides is observed, particularly, in the equatorial MLT region. Thus, the main reason for the difference in the wave forcing in the MLT region is that the amplitude of tides in the analysis becomes realistic in the new analysis (Fig. 3).

Below the weak wind layer at heights of 75–90 km in 30° S–80° S in the mesosphere, the sign of the E-P flux divergence and the direction of the vertical component of the E-P flux are opposite for each analysis (Fig. 6a and 6b). It is

found that the difference is large for small-scale waves with zonal wavenumbers larger than seven (not shown). In the KSMW analysis, these small-scale waves are mainly attributable to the upward propagating spurious waves. Thus, the suppression of the spurious waves by adapting the IAU filtering is likely responsible for the difference in the wave forcing in the southern mesosphere. Besides, it is considered that the increase in the number of satellite observations and the tuning of the horizontal diffusion in the new assimilation system also improve the representation of small-scale waves in the MLT





region. It is interesting to note that, around the weak wind layer at 40° S, the difference of the zonal mean zonal wind between the KSMW analysis and the new analysis is smaller than 10 m s$^{-1}$ in spite of the significant difference in the wave forcing.

## 4 Summary and concluding remarks

The data assimilation system for the height region from the surface to the lower thermosphere developed by KSMW20 was

updated to better represent disturbances with wave periods shorter than one day such as atmospheric tides, which have large amplitudes and may induce significant wave forcing in the MLT region. In the present study, (i) the IAU filtering was introduced, (ii) the horizontal diffusion (hyperdiffusion) was modified, and (iii) observations from the SABER and SSMIS were also assimilated. The validity was confirmed by comparison with independent data, i.e., horizontal winds from meteor radar observations at Longyearbyen in the Arctic, at Kototabang in the equatorial region, and at Davis Station in the

Antarctic. Details of the results are summarized as follows:

- Large amplitudes of disturbances with small horizontal scales of about 1000 km in the MLT region observed in the KSMW analysis were attributable to the analysis increments, because these disturbances were not observed in the free-run simulation with the same initial condition. Thus, first, the IAU filtering was introduced. This could efficiently reduce these spurious waves.

- Next, to obtain realistic tidal wave amplitudes in the MLT region, a horizontal hyper diffusion of the eighth order was introduced so that only small-scale fluctuations were effectively diffused. The tidal wave amplitudes in the time series of the zonal wind from the new analysis became reasonable but still were somewhat larger than those from the meteor radar observations.

- Finally, the SABER temperature retrieval and SSMIS brightness temperature retrieval in the stratosphere and

mesosphere were also assimilated. The correlation between the horizontal wind time series estimated by the new assimilation method and those from meteor radar observations was higher. The wave amplitudes of the winds in the upper mesosphere also were closer to those from the radar observations.

Assimilation of both sun-synchronous and non-sun-synchronous satellite observations is important for better representation of the zonal mean field as well as for short-period fluctuations such as tides. In fact, at Longyearbyen, the

quasi-half-day fluctuations with large amplitudes observed by the meteor radar are well reproduced in the present analysis (Fig. 2m). The correlation between the radar observations and the products of our assimilation system using both types of satellite observations is high (around 0.64), but not very high. This may be partly because of the existence of unresolved gravity waves in our model. Shibuya et al. (2017) simulated the 12-h period disturbances that are observed by a mesosphere-stratosphere-troposphere radar (a VHF clear-air Doppler radar) at Syowa Station in the Antarctic in winter using a high-

resolution high-top GCM and showed that they are due to inertia-gravity waves with horizontal wavelengths of 1000–2000



km. Such relatively short horizontal waves are hardly simulated by the numerical model with T42 used in our assimilation system.

It is worth noting that the mean winds are well reproduced even by our previous assimilation system (KSMW20), although a slight bias remained. However, the wave forcing properties have been largely modified. For example, around the

weak wind layer in the mesosphere of the Southern Hemisphere, the sign of E-P flux divergence and the direction of the vertical component of the E-P flux are opposite between the previous and the new analyses. These are mainly attributable to the reduction of spurious small-scale waves caused by assimilation increments. In the equatorial MLT region, the deceleration of the westerly wind associated with the E-P flux convergence in the present analysis is also modified. This is because of better representation of tidal waves there. Such better reproduction of waves by the new assimilation allows us to

study the momentum budget in the MLT region quantitatively.

Our data assimilation system employs the 4D-LETKF method. Therefore, the computational cost is low. We plan to carry out a long-period analysis using the new assimilation system updated in the present study, over about 16 years from the start of the MLS observations in August 2004 to the present, to examine the dynamics of the MLT variations at time scales from days to years.

**Code and data availability**

For legal reasons, the source code for the forecast model, data assimilation module, and run scripts cannot be publicly released. They have been made available to the editor and reviewers, and are available to anyone by contacting the corresponding author. The copyright of the original code for LETKF belongs to Takemasa Miyoshi, and it can be accessed from https://github.com/takemasa-miyoshi/letkf (last access: 26 June 2020, Miyoshi, 2016). Meteor radar data from

Kototabang are available at the Inter-university Upper atmosphere Global Observation NETwork (IUGONET) site (http://database.rish.kyoto-u.ac.jp/arch/iugonet/mwr_ktb/index_mwr_ktb.html, last access: 25 January 2021, IUGONET, 2016). Meteor radar data from Longyearbyen are available on request from the National Institute of Polar Research by contacting Masaki Tsutsumi (tutumi@nipr.ac.jp). Meteor radar data from Davis (Murphy, 2017) are available online. NCEP PREPBUFR data are also available online. Aura MLS data (Schwarts et al., 2015), which are compiled and archived by

NASA, were also used for the data assimilation. SABER data can be downloaded from the FTP site at ftp://saber.gats-inc.com/Version2_0/Level2A/ (last access: 19 October 2020). SSMIS data can be downloaded from https://www.avl.class.noaa.gov/saa/products/search?&datatype_family=DMSP (last access: 25 January 2021).

**Author contributions**

DK, MK, and KS designed the experiments, and DK carried them out. SW developed the forecast model code. DK, MK, and

KS prepared the paper with contributions from all the coauthors.





**Competing interests**

The authors declare that they have no conflict of interest.

**Acknowledgements**

We greatly appreciate Masaki Tsutsumi and Chris Hall for providing the meteor radar data from Longyearbyen. Operation of

the Davis meteor radar was supported through Australian Antarctic Science project number 4025 and we deeply thank the project CI Damian Murphy for providing the data. DK thanks Kazuyuki Miyazaki for his continuous encouragement. The data assimilation experiments were performed using the Japan Agency for Marine-Earth Science and Technology (JAMSTEC) Data Analyzer (DA) system. The figures were produced by the GFD-DENNOU Library.

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





**Table 1: Parameters obtained in Koshin et al. (2020) for the assimilation system of the whole neutral atmosphere.**

|  | Data assimilation setting for the middle atmosphere |
|---|---|
| Gross check coefficient | 20 |
| Localization length | 600 km |
| Inflation factor | 15% |
| Assimilation window | 6 hours |





**Table 2: Improvements used in the experiments.**

| Experiment | IAU | Diffusion | SABER | SSMIS |
|------------|-----|-----------|-------|-------|
| KSMW (Ctrl) |   | Fourth order |   |   |
| I | x | Fourth order |   |   |
| II | x | Eighth order |   |   |
| III | x | Eighth order | x |   |
| IV (new) | x | Eighth order | x | x |






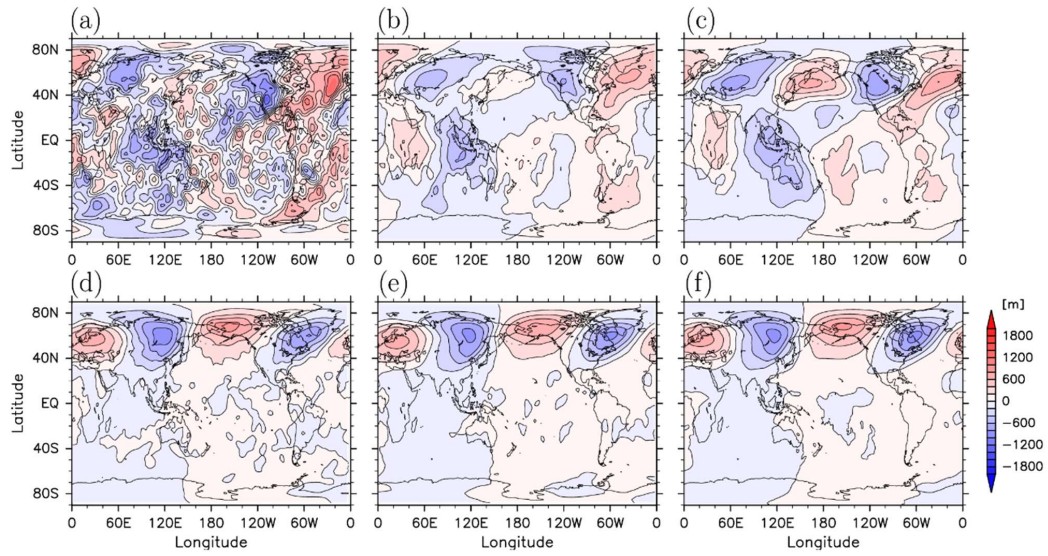

**Figure 1: (a–c) The longitude-latitude sections of the geopotential height anomaly from the zonal mean in 00UTC on 20 January 2017 at 0.1 hPa from (a) the KSMW analysis, (b) the analysis with the IAU, and (c) the MERRA-2 reanalysis. The contour intervals are 200 m. (d–f) Same as (a–c) but for 10 hPa.**



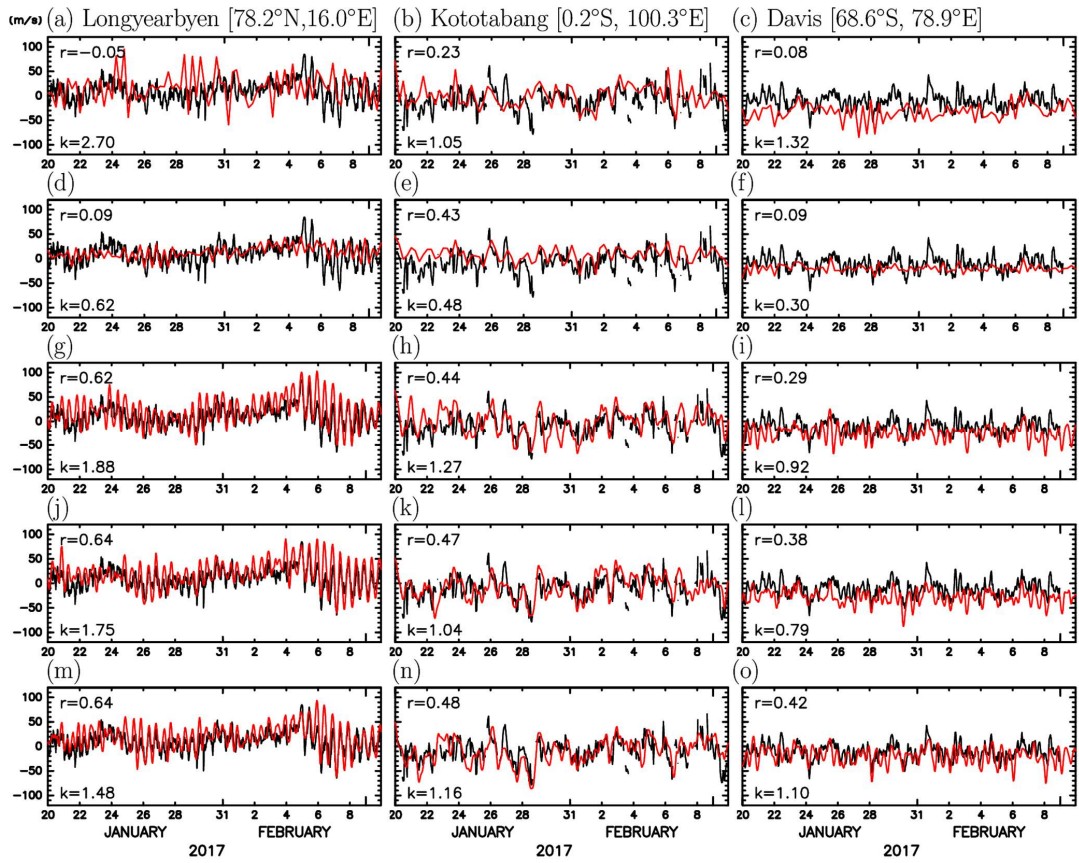

Figure 2: The time series of the zonal wind from analysis (red curves) and observations (black curves) by meteor radars at Longyearbyen in the Arctic (left column), Kototabang near the Equator (middle column), and Davis Station in the Antarctic (right column). The results for (top to bottom) the KSMW analysis, Expt. I, Expt. II, Expt. III, and Expt. IV. Each panel lists the correlation coefficient r and ratio of the variances k between the analysis and corresponding meteor radar wind time series.



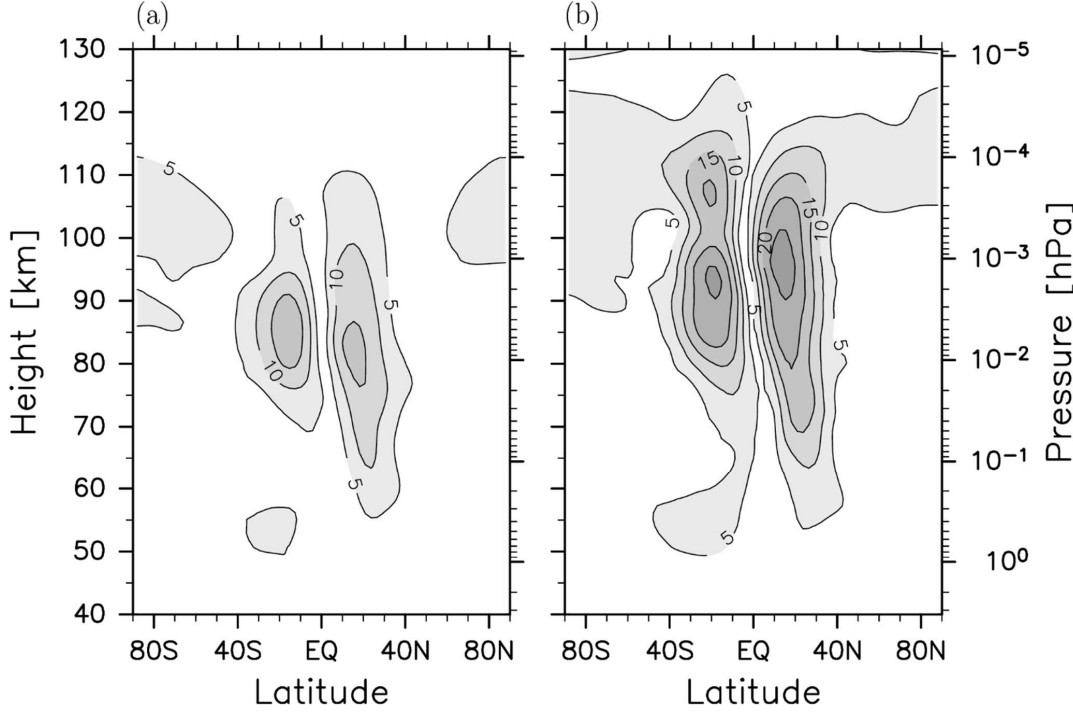

555 **Figure 3: Amplitudes of the migrating diurnal tide in the meridional wind from the free-run simulation with (a) the fourth-order horizontal diffusion and (b) the eighth-order horizontal diffusion for the time period from 15 January to 20 February 2017. The contour interval is 5 m s⁻¹.**

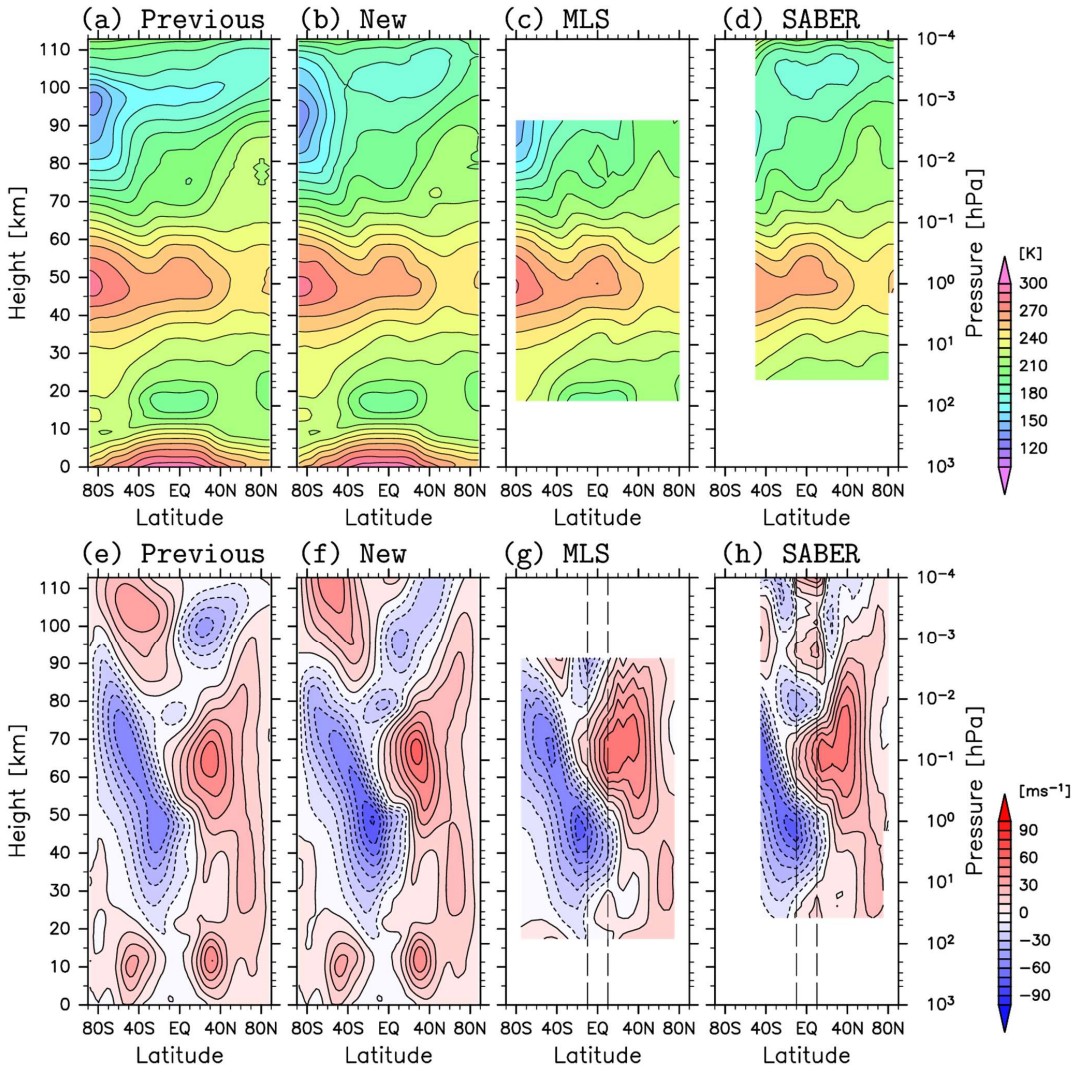

**Figure 4: (a–d) Zonal mean temperature and (e–h) zonal wind for the time period from 15 January to 20 February 2017. The results in panels (a) and (e) are from the KSMW analysis, (b) and (f) are from the new analysis (Expt. IV), (c) and (g) are from the MLS observation, and (d) and (h) are from the SABER observation. The gradient winds from the satellite observation (g and h) at 10° S–10° N are linearly interpolated. Contour intervals are 10 K for (a–d), 10 m s$^{-1}$ for (e–h).**



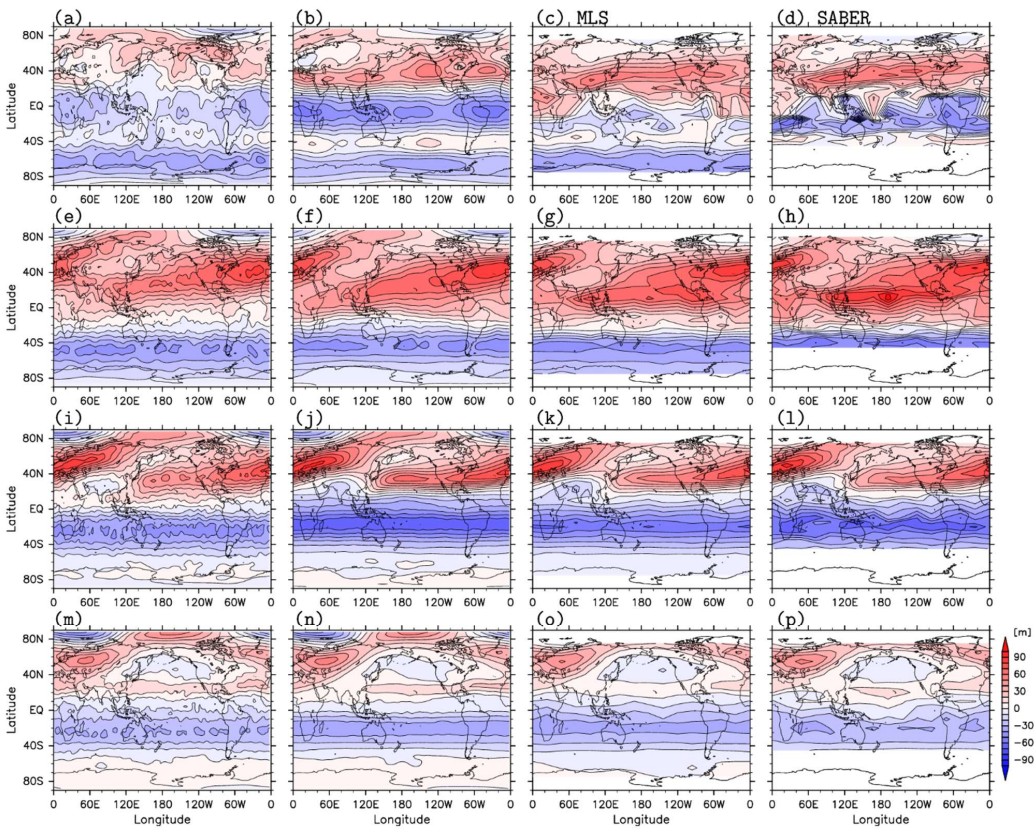

**Figure 5: The longitude-latitude sections of zonal wind from the analyses and zonal gradient wind from the satellite observations,**
**which are averaged for the time period of 10–20 February 2017 at 0.01 hPa (a–d), 0.1 hPa (e–h), 1 hPa (i–l), and 10 hPa (m–p). The**
**results in panels (a), (e), (i), and (m) are from the KSMW analysis, (b), (f), (j), and (n) are from the new analysis (Expt. IV), (c), (g),**
**(k), and (o) are from the MLS observation, and (d), (h), (l), and (p) are from the SABER observation. The gradient winds from the**
**satellite observation (right two columns) at 10° S–10° N are linearly interpolated. Contour intervals are 10 m s$^{-1}$.**





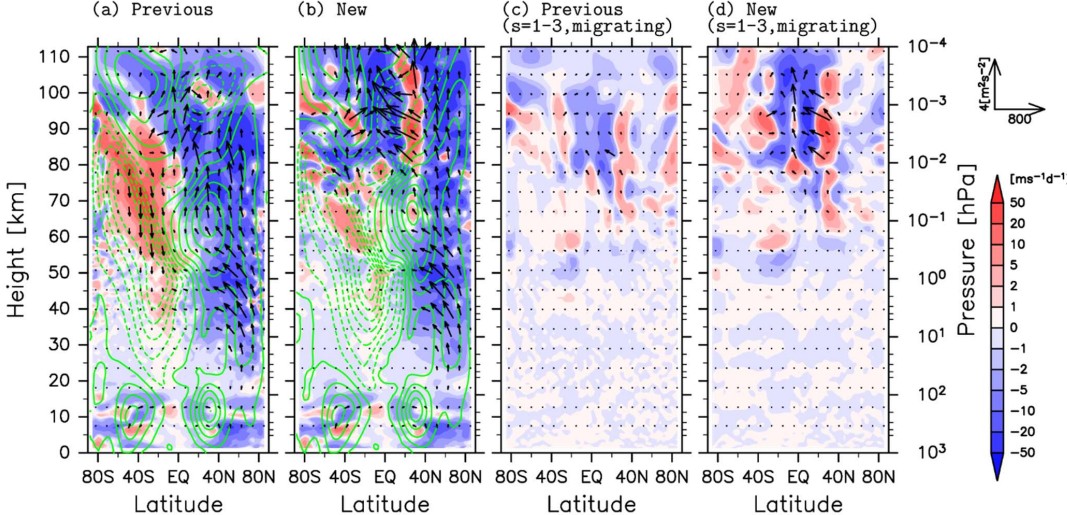

570 **Figure 6: The zonal mean zonal wind (green contours), E-P flux (arrows), and E-P flux divergence (colors) averaged for the period of 15 January to 20 February 2017. The E-P flux of panel (a) is the result of the all waves for the KSMW analysis [b for the new (Expt. IV) analysis], and that of (c) is the result of the migrating large-scale (wavenumber $s < 4$) for the KSMW analysis [(d) for the new (Expt. IV) analysis]. Contour intervals are 10 m s$^{-1}$.**