# Peer review of "An update on the 4D-LETKF data assimilation system for the whole neutral atmosphere"

_Geoscientific Model Development, 2020_

## Author Response (AR1)

Referee #1,

  The authors greatly appreciate his/her critical reading of our manuscript and constructive comments. We have revised the manuscript as much as possible following his/her comments. Our response to each comment is in the following.

Response to comments:

Major comments:

>1. My main comment concerns the discussion of the second model change with the introduction of the new diffusion model. In the numerical results this change leads to improvements compared to just using the IAU filter alone. However, I do not feel this algorithmic change is motivated or explained adequately in the present manuscript. This can easily be changed some of the existing content, and including some additional discussion in the introduction. More specifically:

a) The use of the IAU is discussed in detail and at length in the introduction. However the use of the new hyperdiffusion parameter is only introduced in Section 3.2. It is not clear whether this is a standard tuning approach in the KSMW system, or whether this is a new approach from the authors. A couple of additional sentences in the introduction discussing this aspect of the specific model, and what prior work has been done to select this diffusion parameter would clarify whether this is straightforward tuning, or a more conceptual change.

The introduction of the eighth order hyperdiffuion is also a new approach in this study. It turned out that the fourth order diffusion employed by KSMW20 unrealistically reduces the amplitudes of tidal waves, which are important in the MLT region. This problem could not be solved with the fourth order diffusion by changing the coefficient in the vertical. Thus, we changed the order of the diffusion from fourth to eighth. This drastic change in the diffusion allows us to reproduce realistic amplitudes of the tides. The first, second, third, and fourth sentences of the eighth paragraph in the Section 1 have been added in the introduction.

b) A new subsection should be introduced in Section 2 which discusses the proposed changes to the hyperdiffusion (between Sections 2.2 and 2.3 in the current manuscript). This can be done by moving existing lines 245 - 256 to the new subsection. As this paragraph concerns the free running model and motivates the new choice of diffusion, it would fit better in the methodology section than in the experimental results. This would also improve the flow of section 3.2. I also suggest an additional sentence which justifies explicitly why an eighth order approach is more appropriate here than the existing fourth order approach.

The form of the diffusion in numerical models affect the amplitude of the waves stronger for larger waves. In other words, a higher-order diffusion form can weaken the diffusion for large-scale waves and strengthen that for small-scale waves. The second and third sentences of the second paragraph in

the section 2.3 have been added. The third paragraph and the first, second, third, fourth, fifth, and sixth sentences of the fourth paragraph of section 3.2 have been moved to the new subsection (section 2.3), following the reviewer's comment,

>2. I also felt that the authors could emphasise the new contribution of this paper more. This will be helped by the suggestions above, but being more explicit in the conclusion (and possibly the introduction) about the significance of each of the changes would highlight the novelty of the overall paper.

The first, second, and third paragraphs of section 4 have been revised to emphasize the improvement in this paper.

>3. There were some inconsistencies in notation. The author introduces Ctrl, Exp I, Exp II, Exp III and Exp IV/New in Table 2, but these are not consistently used throughout the manuscript and figures. I suggest replacing Exp IV with New, and including this notation in Figure captions (in addition to the more descriptive current captions). E.g. in Figure 1 replace with "(a) the KSMW analysis (Ctrl), (b) the analysis with the IAU (Exp I)". Similarly in the caption of Figure 4 plus the subplot titles ("Previous" should be "Ctrl")

We appreciate the reviewer's indication. The notation has been revised to use "Ctrl", "Expt. II", "Expt. III", and "New", explicitly.

**Minor comments and typographical errors:**

>1. L12 replace "used for the middle atmospheric" with "used for middle atmospheric"

The expression has been revised following the reviewer's comment.

>2. L26 replace "There also make the dynamics" with "These also make the dynamics"

The expression has been revised following the reviewer's comment.

>3. L29 replace "predominant" with important or dominant

The expression has been revised following the reviewer's comment.

>4. L41 replace "cause by the primary" with "caused by the primary"

The expression has been revised following the reviewer's comment.

>5. The introduction is quite long and there is some repetition e.g. L47 about GCMs not including MLT.

Following the reviewer's comment, the sentence on ll.46–49 has been revised.

>6. L62 "meteor radar observations for several years": does this mean "over several real-time years" or "for several model years".

The former is correct. The sentence on l.62 has been revised.

>7. L69 replace "using the JAGUAR" with "using JAGUAR" or "using the JAGUAR system"

The expression has been revised following the reviewer's comment.

>8. L82: include a comma after "are assimilated,"

The expression has been revised following the reviewer's comment.

>9. L86: replace "A forecast initialised by analysis" with "A forecast initialised with an analysis"

The expression has been revised following the reviewer's comment.

>10. L100: these is an unnecessary linebreak

The expression has been revised following the reviewer's comment.

>11. L122: This sentence is confusing - maybe split into two sentences or reorder.

Following the reviewer's comment, the sentence on the beginning of the section 2.1 has been revised.

>12. L137; replace "parameters by KSMW20" with "parameters from KSMW20"

The expression has been revised following the reviewer's comment.

>13. L150: replace "consumes the calculation time about 10 times as much" with "requires 10 times the amount of computation time"

The expression has been revised following the reviewer's comment.

>14. L153: replace "so as to express" with "in order to express"

The expression has been revised following the reviewer's comment.

>15. L164: replace "Similar" with "Similarly"

The expression has been revised following the reviewer's comment.

>16. L192: I believe this line refers to a missing table which describes the independent observation data. Table 2 in the manuscript describes the different assimilation experiments, and I don't see why that is relevant in this sentence.

We appreciate the reviewer's indication. It is true that the sentence referred to a missing table. This sentence has been removed.

>17. L200: "are called the" instead of "are called as the"

The expression has been revised following the reviewer's comment.

>18. L204: "reproduction" instead of "reproductivity"

The expression has been revised following the reviewer's comment.

>19. L216: replace "inherently" with "by definition"

The expression has been revised following the reviewer's comment.

>20. L222: Could the vertical changes throughout the atmosphere be illustrated in a meaningful way?

We have added a new figure that compares the vertical profile of the amplitude of small-scale waves for the experiments with and without the IAU.

>21. L261: Include an extra sentence to make it clear that 1 is good.

Following the reviewer's comment, a sentence has been added.

>22. L267: "Finally" instead of "Lastly"

The expression has been revised following the reviewer's comment.

>23. L270-271: The changes to L and K are small here, and the description makes them sound more significant. Also this sentence could be re-ordered to reduce the number of parentheses.

Following the reviewer's comment, the second sentence on section 3.3 has been revised.

>24. L277: "there are no SABER" instead of "there is no SABER"

The expression has been revised following the reviewer's comment.

>25. L276 - 280: I found the ordering here confusing. I suggest rearranging to 1) inclusion of new observations is important (currently the final sentence). 2) Improvements for SABER correlation vs variances 3) At Davis SSMIS has more impact due to Southern Hemisphere differences.

Following the reviewer's comment, the sentences at the end of section 3.3 have been revised.

>26. L296: It would be less confusing if the ordering were consistent between the figure and the discussion (e.g. from the bottom of the atmosphere to the top).

Following the reviewer's comment, the sentence at the second paragraph of section 3.4 has been revised.

>27. L364: replace "allows us to" with "will allow us to" - I don't think this work has been done in the current manuscript

The expression has been revised following the reviewer's comment.

>28. L451: There should be a linebreak after 2020 for the new citation

The expression has been revised following the reviewer's comment.

>29. Table 2: The caption could be more informative: e.g. notation for the different experiments considered in this study

Following the reviewer's comment, the caption has been revised.

>30. Figure 4: L561 observation should be observations (two occurrences)

The expression has been revised following the reviewer's comment.

>31. Figure 4: L565: "time period from" not "time period of"

The expression has been revised following the reviewer's comment.

Referee #2,

The authors greatly appreciate his/her critical reading of our manuscript and constructive comments. We have revised the manuscript as much as possible following his/her comments. Our response to each comment is in the following.

Response to comments:

>L23. 'The' does it refer to one GCM in particular? Or to all in general?

The latter is correct. The sentence on l.23 has been revised.

>L26. Last sentence in line is grammatically incorrect.

Following the reviewer's comment, the sentence on l.26 has been revised.

>L30. The first sentence in this paragraph is quite long. Consider splitting in 2 shorter sentences.

Following the reviewer's comment, the sentence on ll.28–31 has been revised.

>L48. Maybe a reference for these GCMs would be useful

Following the reviewer's comment, a reference has been added.

>L85. The explanation about the generation of spurious waves by large analysis increments can benefit by using a figure, or referring to one in some of the existing papers dealing with this topic.

Following the reviewer's comment, the sentences on ll. 83–85 have been revised.

>L200. To refer sometimes to KSMW20 and sometimes to KSMW (without the year) can be confusing. I get that sometimes you refer the paper itself, and sometimes the actual system and analysis, but I think readers will find this cumbersome.

Following the reviewer's comment, the notation has been revised. Koshin et al. (2020) is called KSMW20, the assimilation system is called KSMW20 system, and the analysis is called KSMW20 analysis.

>L210. It may be worth to show the anomaly coming solely from the background so that one can notice the effect of the observations.

The description here may be misleading. The anomaly does not come from the background (i.e., prediction by the model) but reflects artificial small-scale disturbances in the analysis. We wanted to reduce the artificial small-scale disturbances caused by the adjustment of locally large increments, which is originating from the difference between the background and the observation. The first, second, and third sentences have been added at the beginning of section 3.1.

>L220. It is quite interesting to note that the amplitude of small waves is larger in high latitudes. Can this be related to any other known facts of the MLT region. You mention a bias of the model, maybe you can provide a reference about these previously observed facts.

I think that the reviewer misread "latitude" as "altitude". The amplitude of small waves seems to be larger in high altitudes, and not in high latitudes.

>L227. Here again, I feel like the use of KSMW20 and the KSMW analysis is redundant. It can just

be 'their' analysis.

The expression has been revised following the reviewer's comment.

>L280. New analysis does not say much. I would recommend something that indicates an incremental analysis update has been used.

The description of the experiments at the beginning of the section 3.3 has been revised to show that the IAU was included.